# Twinning-assisted dynamic adjustment of grain boundary mobility

Qishan Huang [1,2,5], Qi Zhu [2,5], Yingbin Chen[2,5], Mingyu Gong[3], Jixue Li[2], Ze Zhang[2], Wei Yang[1], Jian Wang[3✉], Haofei Zhou [1✉] & Jiangwei Wang [2,4✉]

Grain boundary (GB) plasticity dominates the mechanical behaviours of nanocrystalline materials. Under mechanical loading, GB configuration and its local deformation geometry change dynamically with the deformation; the dynamic variation of GB deformability, however, remains largely elusive, especially regarding its relation with the frequently-observed GB-associated deformation twins in nanocrystalline materials. Attention here is focused on the GB dynamics in metallic nanocrystals, by means of well-designed in situ nanomechanical testing integrated with molecular dynamics simulations. GBs with low mobility are found to dynamically adjust their configurations and local deformation geometries via crystallographic twinning, which instantly changes the GB dynamics and enhances the GB mobility. This self-adjust twin-assisted GB dynamics is found common in a wide range of face-centred cubic nanocrystalline metals under different deformation conditions. These findings enrich our understanding of GB-mediated plasticity, especially the dynamic behaviour of GBs, and bear practical implication for developing high performance nanocrystalline materials through interface engineering.

[1] Center for X-Mechanics and State Key Laboratory of Fluid Power and Mechatronic Systems, Department of Engineering Mechanics, Zhejiang University, 310027 Hangzhou, China. [2] Center of Electron Microscopy and State Key Laboratory of Silicon Materials, School of Materials Science and Engineering, Zhejiang University, 310027 Hangzhou, China. [3] Mechanical and Materials Engineering, University of Nebraska-Lincoln, Lincoln, NE 68583, USA. [4] Wenzhou Key Laboratory of Novel Optoelectronic and Nano Materials, Institute of Wenzhou, Zhejiang University, 325006 Wenzhou, China. [5] These authors contributed equally: Qishan Huang, Qi Zhu, Yingbin Chen. ✉email: jianwang@unl.edu; haofei_zhou@zju.edu.cn; jiangwei_wang@zju.edu.cn

Nanocrystalline materials possess a large volume fraction of grain boundaries (GBs), which can substantially modify their physical, mechanical and chemical properties in comparison with the coarse-grained polycrystalline counterparts[1–3]. However, nanocrystalline materials have long been suffering from their poor ductility and strain softening[4], due to the plastic instability and thereby premature necking induced by GB deformation[5]. Numerous studies have reported the GB-dominated plasticity via GB migration[6], GB sliding[7,8], and grain rotation/coalescence[9] in nanocrystalline materials. Nonetheless, the dynamic deformability of GBs upon mechanical loading has been largely overlooked in these models. Generally, GB mobility depends not only on the intrinsic GB geometry and atomic structure (such as curvature, misorientation, inclination, impurities, etc.)[10–13], but also on the local stress condition and thermo-mechanical loading history[6,14,15]. During plastic deformation, GB configuration and deformation geometry evolve dynamically with the emission or absorption of defects at GBs. For instance, GB-mediated deformation twinning changes the GB structure significantly, resulting in an instant modification of GB dynamics[16,17]. The variation of GB dynamics should bear an impact on the GB-dominated deformation and even plastic instability. A systematic exploration of the dynamic deformability of GBs is thus critical for a thorough understanding of the plastic instability of nanocrystalline materials, as well as on the application of GB engineering in nanomaterials design.

In nanocrystalline materials, the twinning-modified GB dynamics should become pronounced due to an increased tendency for deformation twinning, even in metals with high stacking fault energies[18,19], where GBs act as the effective twin nucleation sites. In previous studies, deformation twinning is simply deemed as an intragranular deformation mode that is important for the mechanical properties and plasticity of nanocrystalline materials[20,21]. Given that deformation twinning is a reorientation process that not only changes the local lattice orientation dynamically but also tunes the GB structure and thereby GB kinetics simultaneously, the GB-correlated deformation twinning may impose critical influences on GB-dominated deformation[16], rather than simply acting as twin nucleation sites. Such dynamic GB behaviour resembles the common approach of GB engineering, where vast Σ3 boundaries (in the form of annealing twins) were introduced into the polycrystalline materials to regenerate the overall GB networks into a crack-resistant interconnection with a higher portion of special GBs[22]. Hence, the twinning-modified GB structure and geometry underscore the intrinsic GB dynamics during plastic deformation, which can greatly tune the GB mobility and facilitate GB plasticity, as exemplified by twinning-correlated nanograin coarsening or coalescence in face-centred cubic (FCC) nanocrystalline metals under uniaxial tensile loading[16], cyclic loading[23,24] or creep test[25]. A comprehensive understanding of the atomistic mechanism underlying twinning-assisted GB motion is of general significance for the plasticity and GB engineering of nanocrystalline materials, which, however, remains largely elusive due to the lack of quantitative experimental studies.

Here, the dynamically adjusted deformability of GBs under mechanical loading has been unambiguously demonstrated in metallic bicrystals using integrated in situ high-resolution transmission electron microscope (HRTEM) nanomechanical testing and atomistic simulations. The as-fabricated high-angle GBs (HAGBs) with relatively low mobility instantly tune their local lattice orientations and atomic configurations via a GB-stimulated twinning process. Such self-driven dynamic adjustment of GB structure changes the GB dynamics with enhanced GB mobility, leading to an increased GB migration rate in subsequent deformation. A geometry-based model was further proposed to quantitatively describe the dependence of self-driven dynamic

adjustment of GBs on GB misorientation and inclination, by considering the resolved shear stresses on twinning and slip systems. This twinning-assisted adjustment of GB dynamics can well explain the GB-associated twins in a wide range of FCC nanocrystalline metals, offering critical insights into GB-dominated plasticity for GB engineering in nanomaterial design.

## Results

**Twinning-assisted dynamic adjustment of GB structure and deformability.** Nanoscale Au bicrystals with designed GB structures provide a model system to study the dynamic deformability of GBs. Figure 1a shows an as-fabricated Au bicrystal containing a 23° [1$\bar{1}$0] tilt GB, as confirmed by the fast Fourier transform (FFT) pattern in Fig. 1g. The grains on the left and right sides of the GB are denoted as G1 and G2, respectively. Accordingly, the GB between G1 and G2 is denoted as GB$_{1–2}$, with the corresponding atomistic structure shown in Fig. 1h. Atomistic observation indicates that GB$_{1–2}$ contains a few pre-existing GB facets and a nanograin (G3) with a diameter below 2 nm. Subsequently, a tensile loading was applied on this Au nanowire along its axial direction (with an inclination of ~12° to the (002) plane of G2) at a constant rate of ~0.005 nm s$^{-1}$.

Upon tensile loading, the deformation of this Au bicrystal was accommodated by extensive GB migration towards the right, as presented in Fig. 1b–f and Supplementary Movie 1. To quantify the GB dynamics under tensile testing, the cumulative GB migration distance was plotted as a function of the loading time (Fig. 1k). At the beginning, GB$_{1–2}$ migrated rightward via the lateral motion of pre-existing GB facets (Fig. 1a, b), resulting in the growth of G1. An average GB migration rate of ~0.43 Å s$^{-1}$ was derived by calculating the tangential slope of this curve during this stage (Fig. 1k). Subsequently, a deformation twin was nucleated from the intersection between GB$_{1–2}$ and the bottom surface of the bicrystal (Fig. 1b), which extended transversely along the GB and transformed GB$_{1–2}$ into GB$_{T-2}$ (denoting the newly formed GB between the twin and G2). Associated with the lateral growth and the thickening of the deformation twin were the continuously increased segment of GB$_{T-2}$ and the reduced segment of GB$_{1–2}$, leading to the gradual truncation and complete annihilation of G3 (Fig. 1c and Supplementary Fig. 1). Meanwhile, the pre-existing minor GB facets continued to move along the main GB and were finally annihilated at the upper free surface. With the continuous deformation, the twin boundary (TB) penetrated across the crystal, leading to the full twinning (Fig. 1d). Associated with the deformation twinning, the original GB was completely changed to a 47° [1$\bar{1}$0] tilt GB (Fig. 1e). Namely, the ($\bar{1}\bar{1}$1) plane of the twin was tuned to the direction almost parallel to the (002) plane of G2, with only a small misorientation of ~7° (as confirmed by the FFT pattern in Fig. 1i). The atomistic structure of the GB$_{T-2}$ in Fig. 1j further demonstrates the nearly coherent relation between the (002) and ($\bar{1}\bar{1}$1) planes across GB$_{T-2}$. In subsequent deformation, migration of the newly formed GB$_{T-2}$ (in a disconnection-mediated mode[26]) resulted in a consecutive thickening of the deformation twin (Fig. 1e, f and Supplementary Fig. 1). Associated with this process was a sharp increase of the GB migration rate to ~1.4 Å s$^{-1}$ (Fig. 1k), in comparison to ~0.43 Å s$^{-1}$ of the original GB$_{1–2}$. This quantitative analysis of in situ TEM observations provides solid evidence for twinning-facilitated dynamic adjustment of GB structure, which enhances GB mobility with the deformation. In theory, the set-up asymmetrical tilt high-angle GB$_{1–2}$ with a misorientation 23° (Fig. 1g) was unfavoured for GB migration, according to the traditional shear-coupling model[27]; however, the occurrence of deformation twinning tuned the lattice misorientation (to 47°) across the GB, which greatly promoted the shear-

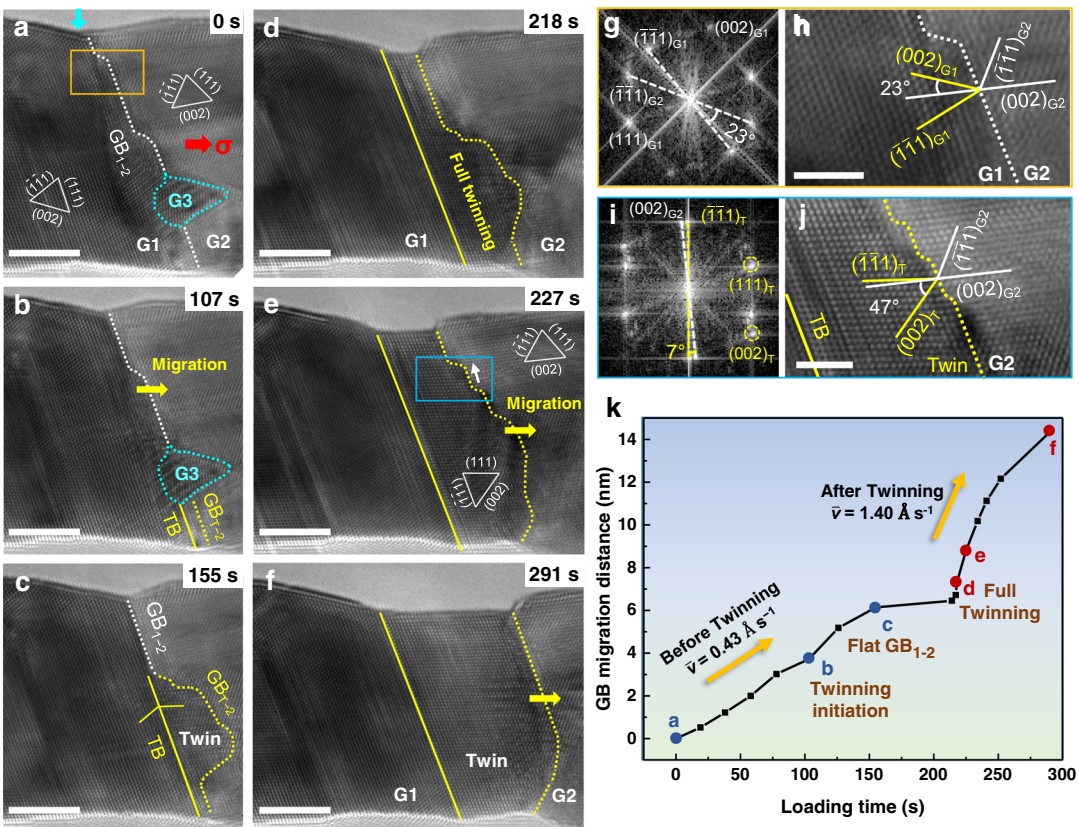

**Fig. 1 Twinning-assisted dynamic adjustment of grain boundary structure and mobility. a** As-fabricated Au bicrystal with a 23° [1$\bar{1}$0] tilt grain boundary (GB) denoted as GB$_{1-2}$ (indicated by the white dotted line). Tensile loading was applied at a constant rate of ~0.005 nm s$^{-1}$ along the axial direction (shown by the red arrow), with a slanted angle of ~12° to the (002) plane of G2. **b** Rightward migration of GB$_{1-2}$ indicated by the yellow arrow. Meanwhile, a deformation twin nucleated at the intersection of GB$_{1-2}$ and the bottom free surface, transforming the local GB segment to GB$_{T-2}$ with a misorientation of 47°, as indicated by the yellow dotted line. **c** Lateral expansion of the GB-emitted deformation twin sandwiched by GB$_{T-2}$ and TB. **d–f** Fully transformed GB$_{T-2}$ with enhanced mobility after deformation twinning, which adjusted from a curved geometry to a smooth one upon further migration with facets lateral motion in a disconnection mode, as indicated by the white arrow in (**e**). **g–j** Fast Fourier transform (FFT) patterns and enlarged images of the GB before (**g** and **h**) and after (**i** and **j**) deformation twinning. The GB regions are marked by the orange and blue boxes in (**a**) and (**e**), respectively. **k** Quantitative measurement of the GB migration distance with increasing loading time. The deformation snapshots **a–f** were highlighted by solid circles in the plot. A static surface step (indicated by the light blue arrow in **a**) was selected as the reference to evaluate the migration distance of the GBs. Scale bars: **a–f** 5 nm; **h**, **j** 2 nm.

coupled GB migration via a disconnection-mediated mechanism, favouring the growth of the deformation twin. These experimental results clearly demonstrate that GB can dynamically tune its deformability via self-driven deformation twinning, which should be strongly correlated with the GB structure and local stress state.

**Microstructural origin of the self-adjustment of GB mobility.** To rationalize the origin of twinning-induced dynamic adjustment of GB mobility, molecular dynamics (MD) simulations were conducted to explore the governing factors from both dynamic and energetic perspectives. Simulation was first performed on the sample with an inclined 23° GB (identical to that of our experiment) under uniaxial tension to validate the twinning-assisted adjustment behaviour of the GB (Fig. 2a). Upon tension, a few embryonic deformation twins were nucleated from the GB, tuning the local GB$_{1-2}$ segments to the GB$_{T-2}$ between the twin and G2 grain (Fig. 2b). These TB segments then interlinked together to promote the growth of the deformation twin, resulting in a perfect coherent TB accompanied by a concomitant GB$_{T-2}$ with a misorientation of 47° (Fig. 2c). To further understand the twinning mechanism, detailed structure evolution of GB$_{1-2}$ was

analysed. The simulated GB$_{1-2}$ contained several pre-existing facets prior to twinning, namely terrace A and facets B, C, mimicking the GB configuration observed in our experiment (Fig. 2d). Such GB facets should result from their anisotropic excessive energies due to the local variation of GB inclination, which thus are rather common among different GBs[28–31]. Under tensile loading, twinning occurred preferentially at the intersections between the main terrace A and facets B and C (Fig. 2e). Accompanied with the twinning process, the minor GB facets B and C propagated separately along the terrace and merged together before they were annihilated at the free surfaces (Fig. 2e). Eventually, the nucleated twin expanded laterally along terrace A, generating a comparatively flat GB$_{T-2}$ and a perfect TB (Fig. 2f).

Given that the deformation processes of twinning and GB migration are dominated by shear stress and the individual role of stress components can always be studied by decoupling, thus, we further study the origin of the twinning-enhanced GB deformability under shear loading by setting up a simulation model with a horizontal 23° GB and a sample size of $10 \times 10 \times 20$ nm$^3$ (Fig. 3a). Periodic boundary conditions were imposed along the GB plane to preclude surface effects. The flat 23° [1$\bar{1}$0] GB$_{1-2}$ was composed of periodically patterned structural units (see the inset in Fig. 3a). The same shearing (Fig. 3a) and tension (Fig. 3b)

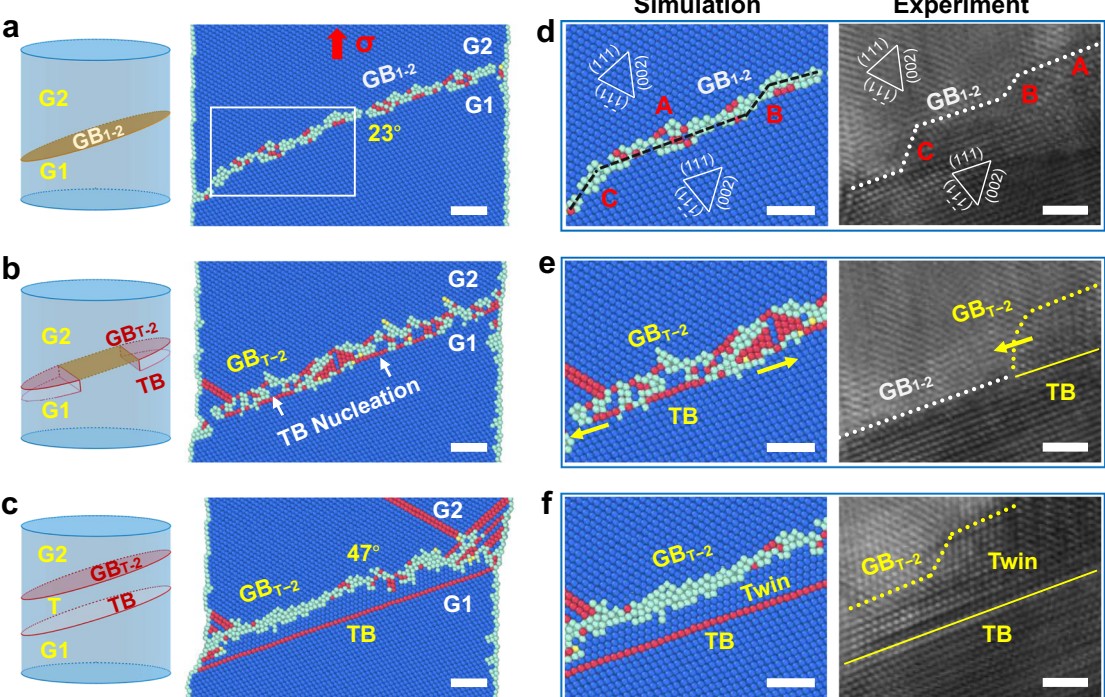

**Fig. 2 Atomistic dynamics of GB-mediated deformation twinning. a** MD simulation performed on an inclined 23° tilt GB under uniaxial tensile loading, clearly validating the twinning-assisted GB structure adjustment . **b** Embryonic deformation twins nucleated from the GB, tuning the local GB$_{1-2}$ segments to GB$_{T-2}$. **c** Growth of the deformation twin via the interlink of TB segments, resulting in a perfect coherent TB accompanied by a concomitant GB$_{T-2}$ with a tilt angle of 47°. Left schematic insets of (**a**–**c**) illustrating the inclined GB configurations in the twinning process. **d**–**f** Atomistic simulation and experiment elucidating the detailed twinning process accompanied by the lateral motion of facets along GBs. **d** The initial GB$_{1-2}$ contained typical GB facets B and C, as well as the main terrace A, which possessed the same misorientation but different inclinations. **e** Preferential twinning occurred at terrace A, while the remaining minor facets B and C moved laterally along the GB. Yellow arrows indicate the direction of twin growth. **f** Embryonic twin expanded along terrace A, resulting in a perfect coherent TB with a comparatively flat concomitant GB$_{T-2}$. Atoms with face-centered cubic (FCC), hexagonal close-packed (HCP) and disordered structures were coloured in blue, red and cyan. All scale bars: 2 nm.

loading rates of $1\,m\,s^{-1}$ were imposed on the sample with directions parallel and perpendicular to the GB, respectively. Under shear loading, deformation instantly induced the rearrangement of local atoms in GB$_{1-2}$. With increasing deformation, embryonic twins formed at some GB segments and local GB$_{1-2}$ segments were tuned into the configuration of GB$_{T-2}$ between the twin and G2 grain (Fig. 3a, $\gamma = 1.33\%$). Subsequently, the remaining segments of GB$_{1-2}$ were transformed into GB$_{T-2}$ via the continuous formation and interlink of atomic twin embryos (Fig. 3a, $\gamma = 1.67\%$, Supplementary Movie 2). These twin segments interlinked together at a shear strain of ~5%, generating a perfect coherent TB with a concomitant GB$_{T-2}$ with a tilt angle of 47° (Fig. 3a, $\gamma = 4.67\%$). Such twinning-assisted GB structure adjustment was associated with decreased GB energy from $816.3\,mJ\,m^{-2}$ for GB$_{1-2}$ to $810.3\,mJ\,m^{-2}$ for GB$_{T-2}$. The resultant GB$_{T-2}$ migrated steadily in subsequent shear loading (Fig. 3a, $\gamma = 33.3\%$), fully consistent with our experimental observations and MD simulations (Figs. 1 and 2). In contrast, tensile loading only induced dislocation nucleation from the GB, with negligible intrinsic GB migration (Fig. 3b). Therefore, simulations combining tension and shear on this model show consistent twinning and GB dynamic adjustment behaviour (Supplementary Fig. 2). These comparisons validate that the twinning-assisted adjustment of GB deformability is a shear-driven process independent of free surfaces[32].

It is known that deformation twinning is a crystallographic reorientation process. In nanocrystalline materials, GB can facilitate deformation twinning via GB decomposition or partial dislocation emission, which modifies the lattice misorientation

across the GB and thus provides sufficient space to dynamically adjust the GB deformability. To quantify the dynamic change of GB deformability, the shear coupling factors of the GBs (defined as $\beta = v_{\parallel}/v_{\perp}$, where $v_{\parallel}$ is the grain translation velocity and $v_{\perp}$ is the GB migration velocity) before and after twinning were calculated by linear fitting of the relationship between the GB migration distance and the shear displacement. Supplementary Fig. 3 illustrates the shear coupling factors for GB$_{1-2}$ and GB$_{T-2}$ obtained from our experimental measurements and MD simulations, which quantitatively shows a lower shear coupling factor for GB$_{T-2}$ than GB$_{1-2}$, indicating enhanced shear deformability of GB$_{T-2}$. Note that these data points from experiments and simulations deviate from the theoretical curve predicted by a previously proposed shear coupling model[11] (see details in Supplementary Discussion 1), due to the influences of temperature, GB geometry and loading rate, etc. The pre-existing defects in real samples may also affect GB migration and thus reduce the shear coupling factor.

Aside from the change of shear coupling factor, the elastic energies stored in the neighbouring grains also change with the twinning-modified GB structures. Under steady mechanical loading, the energy difference across the GB offers a driving force $P$ for GB migration, which becomes more pronounced in the presence of local stress concentration at the GB segments. As shown in Fig. 3c, the difference between the driving forces $P$ for GB$_{T-2}$ and GB$_{1-2}$ becomes more significant with the increasing normal strain $\varepsilon$ (see Supplementary Discussion 2), which partly rationalizes the higher mobility of GB$_{T-2}$ than GB$_{1-2}$ at finite strain. Moreover, the energy barriers of GB migration before and

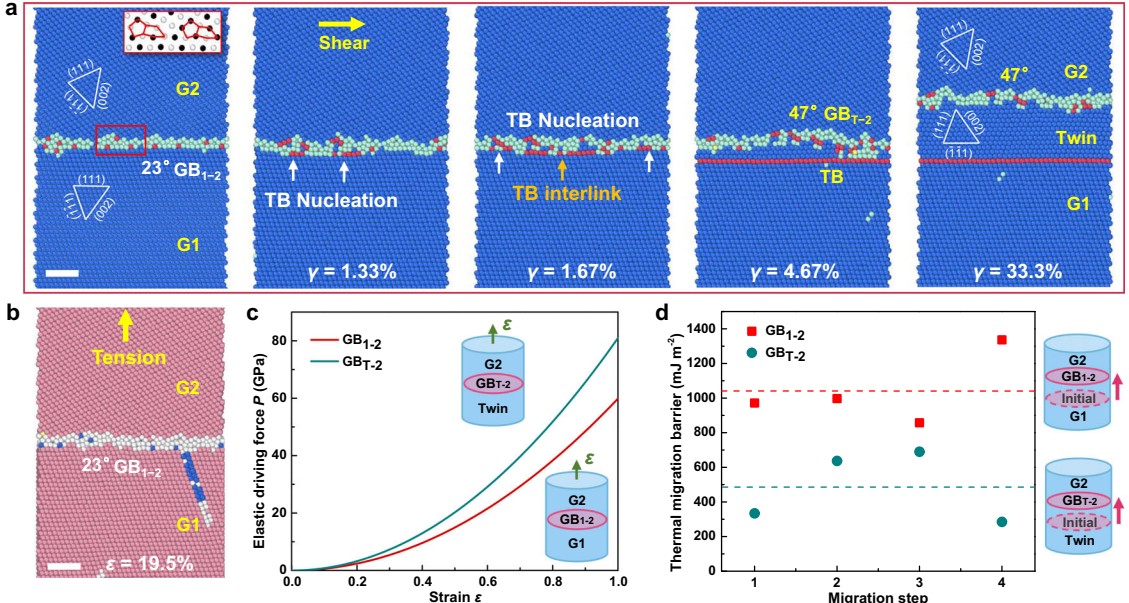

**Fig. 3 Microstructural origin of the self-adjustment of GB mobility. a** MD simulation snapshots demonstrating the atomic-scale structural adjustment during the twinning-assisted GB transformation under shear loading. The Au bicrystal constructed in MD simulations, containing a flat 23° tilt GB. The inset presents a regular structure unit of the GB. Nucleation of embryonic TB segments associated with the local rearrangement of atoms at $GB_{1-2}$, as indicated by the white arrows ($\gamma = 1.33\%$). Interlink of TB segments along the GB with further TB nucleation events ($\gamma = 1.67\%$). Complete transformation from $GB_{1-2}$ to $GB_{T-2}$ via the interlink of twin segments ($\gamma = 4.67\%$). The newly formed $GB_{T-2}$ migrated upward while TB remains static in subsequent shear loading ($\gamma = 33.3\%$). **b** The same bicrystal deformed under axial tensile loading, which is dominated by elastic deformation of the grains and dislocation emission with negligible GB migration. **c** Theoretical elastic driving force $P$ for the migration of $GB_{1-2}$ and $GB_{T-2}$ as a function of the applied normal strain with respect to the GB plane. **d** Comparison between the thermal migration barriers of $GB_{1-2}$ and $GB_{T-2}$ measured from four consecutive migration steps along the GB normal direction, with a migration distance of one atomic layer (approximately 0.3 nm) per unit step. The average values of migration barriers of $GB_{1-2}$ and $GB_{T-2}$ from these migration steps are highlighted by the red and cyan dotted lines, respectively. Atoms with FCC, HCP and disordered structures were coloured in blue, red and cyan in (**a**); and in pink, blue and white in (**b**). All scale bars: 2 nm.

after full twinning have been compared using the nudged elastic band (NEB) method[33], given that GB migration is a thermally activated deformation process. Starting from the initial GB structure, four consecutive migration steps along the GB normal direction have been performed for both $GB_{T-2}$ and $GB_{1-2}$ (see the insets of Fig. 3d). In each unit step, the GB migrated upwards with a specific distance of approximately 0.3 nm (corresponding to one-atomic-layer spacing). The average migration energy barrier of $GB_{T-2}$ (cyan dotted line in Fig. 3d) is much lower than that of $GB_{1-2}$ (red dotted line in Fig. 3d), further supporting the twinning-enhanced GB migration rate in the present study.

**Twinning tendency on the self-driven GB mobility adjustment.** Both in situ observations and atomistic simulations have demonstrated that the GBs in Au nanocrystal can adjust its mobility dynamically by GB-mediated deformation twinning. To validate the generality of this unique GB dynamics, additional MD simulations were carried out on ⟨110⟩ tilt GBs with misorientations ($\theta$) ranging from 10° to 70°, while fixing the lattice orientation of G2 (Fig. 4a). When $\theta$ was lower than 16°, the low-angle GBs (LAGBs) were composed of 1/2 ⟨110⟩ dislocation arrays and deformed by the collective motion of GB dislocations along the slip planes in G1 or G2 (Fig. 4e and Supplementary Fig. 4a), consistent with the theoretical prediction[34] and previous experimental studies[35]. For GBs with misorientations ranging from 16° to 36°, a general self-stimulated structural adjustment by deformation twinning was exhibited, as illustrated by the deformation configurations of different samples in Fig. 4b, c and Supplementary Fig. 4b. Upon loading, each GB decomposed into a new GB and a TB to effectively release the deformation-induced

stress accumulation at GB (Supplementary Fig. 5). After twinning, the newly formed GBs could migrate smoothly in subsequent deformation, contributing to an enhanced GB mobility. When $\theta$ exceeded 36°, the self-driven dynamic GB adjustment was rarely observed (Fig. 4d and Supplementary Fig. 4c). Instead, dislocation slip was readily activated from the GB due to the increased resolved shear stress along the slip plane, as exampled by an example of 50° ⟨110⟩ tilt GB in Fig. 4d.

Such misorientation dependence of twinning-assisted GB deformability is related to both the excess energies and the twinning tendency of GBs. As shown in Fig. 4e, GBs prone to twinning-facilitated migration typically possess relatively high excess energies, which furnishes a driving force for the self-stimulated structural adjustment. It is noticed that some GBs beyond 36° also possess high excess energies, despite their slip-controlled behaviour, which can be explained by the lower twinning tendency. To quantify the twinning tendency towards the self-adjusted GB dynamics, we developed a geometrical-based theoretical model considering the resolved shear stresses on the twinning and slip planes in G1 (see Fig. 4a). According to Fig. 4a-d, GB-facilitated twinning occurs via the motion of twinning partials along (111) slip planes in G1 (marked by "Twinning" in Fig. 4a), while the dislocation emission on ($\bar{1}\bar{1}1$) planes leads to slip-governed GB deformation (marked by "Slip" in Fig. 4a). Thus, the twinning tendency (TT) of a GB is defined as the ratio between the resolved shear stresses on "Twinning" and "Slip" planes, i.e., $TT = \tau_{Twinning}/\tau_{Slip}$, which can be further correlated to the GB misorientation in the expression of:

$$TT = |\cos(2\theta - 46°)/\cos(265° - 2\theta)| \tag{1}$$

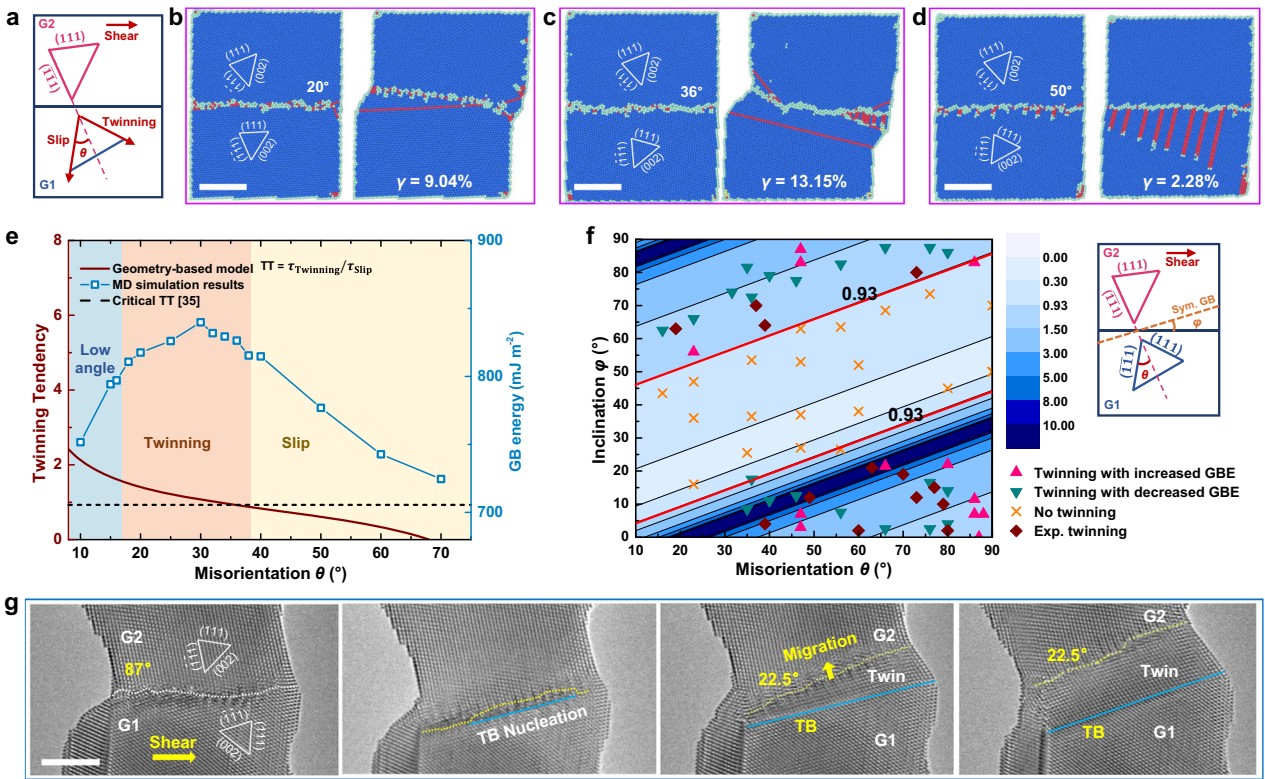

**Fig. 4 Geometry-based twinning tendency model for self-adjust GB deformation. a** Schematic delineating the GB misorientation $\theta$ between G1 and G2, with the twinning and slip planes of G1 labelled. **b, c** Dynamic GB adjustment mediated by deformation twinning of two [1$\bar{1}$0] tilt GBs with $\theta = 20°$ and 36°. **d** Dislocation emission and slip at the GB with $\theta = 50°$. **e** Twinning tendency and GB energy plotted as a function of the GB misorientation. The transitional misorientation for the twinning-slip competition was determined as 35.7°, at which the twinning tendency reaches a critical value of 0.93 (indicated by the black dotted line), confirmed by a series of MD simulation results (indicated by the squared data points). **f** Twinning tendency map considering the collective influences of GB misorientation $\theta$ and inclination $\varphi$, where $\varphi$ is defined as the deviation angle of the actual GB plane from that of the ideally symmetric GB (see the right schematic inset). The critical twinning tendency of 0.93 is marked out in the map, verified by the data points obtained from MD simulations and experiments. The colour bar indicates the value of twinning tendency predicted by the geometry-based model. **g** Deformation snapshots of energetically unfavourable twinning occurred at an 87° [1$\bar{1}$0] tilt GB, leading to the formation of a 22.5° GB$_{T-2}$ and a TB (indicated by the yellow and blue lines, respectively). The newly formed GB$_{T-2}$ migrated smoothly via the collective gliding of GB dislocations in subsequent shear loading. All scale bars: 5 nm.

If TT > TT$_{critical}$ for a given GB, spontaneously adjustment of its deformation dynamics can occur via deformation twinning, and the critical twinning tendency TT$_{critical}$ was determined to be around 0.93 for Au (see details in Supplementary Discussion 3)[35]. This criterion yields a threshold GB misorientation of 35.7° (Fig. 4e), below which twinning dominates over slip-governed GB migration, and vice versa, fully consistent with our simulations.

The twinning tendency in Fig. 4e was obtained with the assumption of a fixed lattice orientation of G2 (Fig. 4a). We further notice that in our TEM observations and atomistic simulations (Fig. 2), twinning was revealed to occur and grow preferentially at the main terrace A, rather than other minor facets B and C, despite the same misorientation $\theta = 23°$, which points out the importance of GB inclination on deformation twinning. To establish the full map of GB self-adjusted dynamic deformation, the effects of varying inclinations were systematically investigated by rotating G2 with respect to the shear loading direction. Accordingly, the geometry-based theoretical model of twinning tendency can be extended as

$$TT = |\cos(109.5° - 2\varphi + \theta)/\cos(250.5° - 2\varphi + \theta)| \quad (2)$$

where $\varphi$ denotes the actual GB inclination from the symmetrical GB plane of two grains (see the right inset of Fig. 4f). Consequently, the terrace A ($\theta = 23°$ and $\varphi = 66°$) is predicted to possess a higher twinning tendency, i.e., TT = 1.28, compared

with the minor facets B ($\theta = 23°$ and $\varphi = 36°$) with TT = 0.53 and C ($\theta = 23°$ and $\varphi = 16°$) with TT = 0.38, respectively, which is consistent with the preferential twinning behaviour of the terrace A (Fig. 2d–f). The contour map in Fig. 4f further clarifies the synergistic effects of GB misorientation and inclination on GB deformation, where the critical value of TT = 0.93 has been highlighted. An additional series of MD simulations have been carried out to verify the model for a wide range of misorientations and inclinations (Supplementary Fig. 6), which have been superimposed onto the contour map in Fig. 4f. In contrast to the energetically favourable formation of twins reported in previous studies[36], the structural adjustment of GBs can either enhance or reduce the GB energy (marked by triangles with different colours in Fig. 4f), indicating that the twinning-assisted GB deformability is insensitive to the GB energy variation in the deformation process. Besides, LAGBs with a misorientation angle <16° are composed of well-aligned GB dislocations with high mobility and thus deform via dislocation slip instead of deformation twinning.

To further validate our geometry-based model, additional testing was performed on a bicrystal with a flat 87° [1$\bar{1}$0] symmetrical tilt GB at a shear rate of ~0.005 nm s$^{-1}$ (Fig. 4g and Supplementary Movie 3). With a theoretical TT of 1.04, the GB is expected to exhibit a self-adjust dynamic behaviour under shear loading. Upon deformation, the GB plane rotated counter-

clockwise for about 13°, aligning the GB parallel to $(\bar{1}\bar{1}1)$ plane of G1; during this process, the initial GB decomposed into a 22.5° GB and a concomitant TB, followed by the constant GB migration in subsequent shear loading (Fig. 4g). MD simulations in Supplementary Fig. 7 and Supplementary Movie 4 confirm the same twinning-assisted GB dynamic adjustment, with a GB energy increase from 785.9 mJ m$^{-2}$ (for 87° GB) to 805.7 mJ m$^{-2}$ (for 22.5° GB). Above MD simulations and in situ experiments have provided a comprehensive understanding of the geometry effect (including both misorientation and inclination) on self-adjusted GB mobility. The self-adjust twinning-assisted GB deformation is general for a wide range of GBs and plays a key role in tuning the GB structure and mobility. The threshold misorientation/inclination varies with the local stress state[37], implying that the dynamic GB deformability adjustment is a stress-dominant process.

## Discussion

GB-dominated plasticity has been widely reported in literature, which exhibited sharply different behaviours under a range of GB configurations/geometries, including misorientation, inclination, curvature etc.[38,39]. However, current theories of GB dynamics mainly rely on analysis of the original GB geometry, which cannot reflect the full landscape of GB plasticity in view of the dynamic evolution of GB structure during deformation. As such, controversial conclusions exist on the mobility of GBs and the stability of nanocrystalline materials[3,40]. For example, several studies revealed that the HAGBs generally possess higher mobility than that of the LAGBs[41,42], while others reported no clear correlation between misorientation and the mobility of HAGBs[40,43]. Our experimental and simulation investigations unambiguously demonstrated that certain GBs in FCC metallic materials can tune their deformability dynamically via a self-driven twinning process. In this process, GB decomposition stimulates the deformation twinning, which, in turn, modifies the GB structure and the associated deformation dynamics, leading to dynamically-adjusted GB mobility.

More importantly, this self-driven twinning-assisted dynamic GB plasticity is independent on surface effects and crystal size, and thus should be quite common in nanocrystalline FCC metals. Fig. 5a–c and d–f further demonstrate accelerated GB migration after twinning in Au films with multiple grains. Under shear loading, the 37° HAGB (Fig. 5a–c) and 19° LAGB (Fig. 5d–f) with low mobility adjusted their structures/misorientations by the nucleation of twin embryos from GBs. With the adjustment of GB structures, these GBs were able to migrate more smoothly, indicating an enhanced deformability. Similarly, in the simulation of a quasi-three-dimensional polycrystalline Au sample, some immobile GBs (e.g., the 87° HAGB in Fig. 5g and Supplementary Fig. 8) preferentially dissociated into a deformation twin and a new GB (22.5° GB) before migration (Fig. 5h, via the gliding of dislocation pairs for this GB), confirming the self-adjusted GB deformability. In these experimental and simulation studies, triple junctions among neighbouring GBs may exert certain pinning effects, inducing a non-uniform migration between different segments of the GBs.

It is noticed that deformation twinning has been frequently observed in as-deformed nanocrystalline FCC metals, most of which were proved to correlate closely with GB deformation[16,17,36]. Some studies have shown that deformation twins in nanocrystalline metals are more likely to nucleate at a lower grain growth rate[44], and the increase of Σ3 GBs can promote grain growth[45]. Similar to the GB dissociation induced twin and thereby self-stimulated adjustment of GB mobility, deformation twins nucleated at other sources can impinge the GBs at the twin front, leading to GB dissociation,

segmentation or partial replacement by incoherent TBs[23,24]. All of these processes could markedly modify the GB structure/misorientation, contributing to the dynamic change of GB mobility[24,44]. These observations suggest that the GB-associated twins (including both nucleation and growth) in nanocrystalline metals, under either mechanical loading or thermal annealing, are more likely to be dominated by the dynamic GB deformation processes, rather than through the migration of TB itself. Taken the ⟨110⟩ tilt GBs as an example (Figs. 1–4 and 5a–h), we have clearly revealed that the self-driven adjustment of GB structure can fundamentally change the GB deformability, which, as the GB migrates, lead to a thickening of the GB-emitted twin among a wide range of GB misorientations and inclinations.

To further confirm this, we have summarized in Fig. 5i the data of GB-mediated deformation twinning from both as-prepared samples (e.g., deposited or after high-pressure torsion) and deformed samples (e.g., subjected to shear/tension/cyclic loading) reported in literature, where the enhanced GB mobility can be traced by identifying the GB structures before and after the twin formation to calculate the corresponding shear coupling factors using MD simulations (see Supplementary Discussion 4 and Supplementary Fig. 9). It is surprising that the GB-emitted twinning and associated self-adjust GB deformation behaviour is universal in nanocrystalline metals with complex GB networks over a wide range of misorientations and grain sizes (especially in the nano-sized regime), where GBs (either LAGBs[46] or HAGBs[47]) were often coupled with TBs, indicating the dynamic transformation between HAGBs and LAGBs. In subsequent loading, the GBs would move or react with other pre-existing defects to further adjust the GB structures[46]. The synergistic motion of GB sliding and migration can also be activated to dynamically adjust the GB networks[48,49]. The GB dislocations generated from such self-driven GB dynamics may further promote grain growth by contributing to the self-driven grain rotation[46]. These dynamic mechanisms call for a rethink of the role of deformation twinning in nanocrystalline materials, especially the ones interlinked with GBs.

In conclusion, the self-driven twinning-assisted dynamic adjustment of GB mobility was systematically investigated using Au bicrystals with HAGBs as model systems. During GB migration, GB structures can frequently experience self-adjustment by shear-dominated twinning, which, in turn, modify the GB structures and effectively promote GB deformability to accommodate migration by producing lower coupling factors and larger driving forces. Such self-adjusted GB migration is a common deformation mode for GBs among a range of FCC nanocrystalline metals under mechanical loading. A GB geometry-based twinning tendency model considering the GB misorientation and inclination was developed to predict the possibility of the twinning-assisted GB adjustment. Our findings provide deep insights into the fundamental understanding of self-adjusted dynamic behaviours of GBs, which predict the stability and evolution of microstructures of metals and alloys with low stacking fault energies, as well as the manipulation of interface dynamics to achieve optimal performance of nanocrystalline materials.

## Methods

**In situ TEM nanomechanical testing.** In situ nanofabrication and tensile testing of the Au bicrystals with [1$\bar{1}$0] tilt GBs were conducted inside an FEI Titan Cs-corrected TEM, equipped with a TEM electrical holder from Beijing PicoFemto Co. In the typical nanofabrication process, an Au rod (99.999 wt.% purity, 0.25 mm in diameter) ordered from Alfa Aesar Inc. was cut by a ProsKit wire cutter to obtain a fresh fracture surface with numerous nanoscale tips and then loaded on the holder as the fixed end; an Au probe on the moving end was driven by the piezo-manipulator to approach the fixed end. At the moment of contact, the Au probe with a pre-applied voltage potential (−1.5 V) and a nanoscale tip on the fracture surface of the Au rod was melted together to form an Au bicrystal with a specific GB structure. Using this method, the orientation and tilt angle of the GB can be tuned by careful manipulation of the Au probe and the application of proper

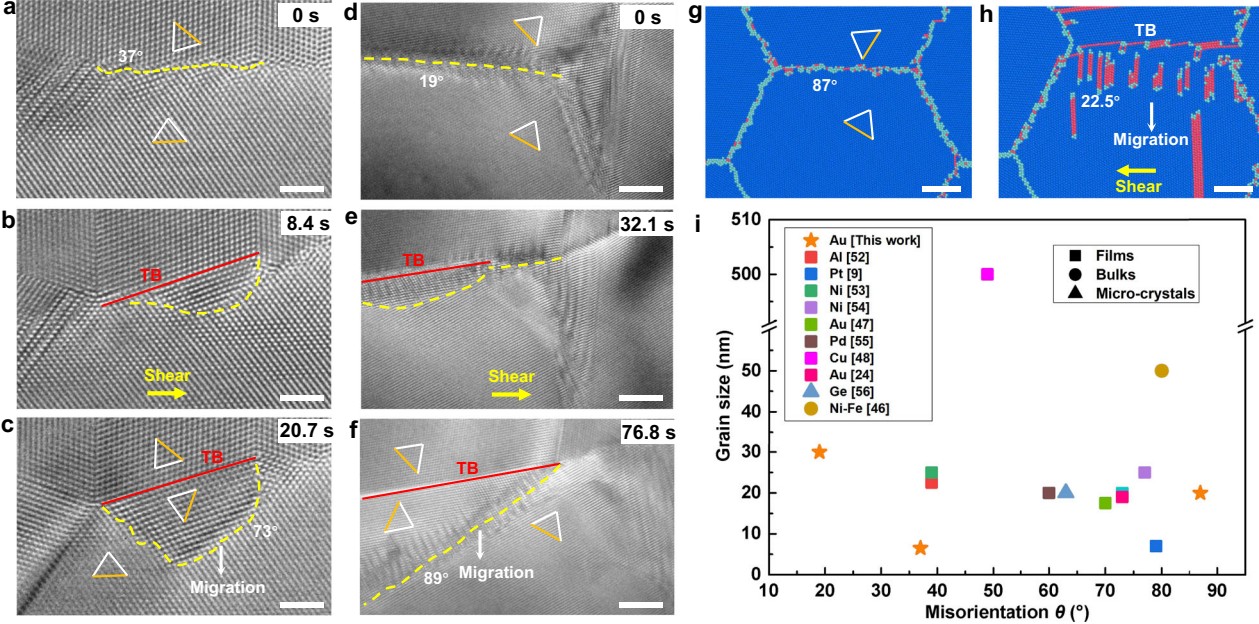

**Fig. 5 Twinning-assisted GB dynamics in nanocrystalline metals. a–c** Self-driven adjustment of a HAGB. **a** A 37° HAGB in nanocrystalline Au film pinned by two GB junctions. **b** TB nucleation accompanied with GB rotation under shear loading. **c** Complete transformation of the GB. **d–f** Self-driven adjustment of a LAGB. **d** A 19° LAGB in nanocrystalline Au film with one end pinned by a triple junction. **e** GB structural transformation assisted by deformation twinning. **f** Migration of the newly formed 89° GB. **g** and **h** MD simulation snapshots of deformation dynamics of an 87° HAGB in a nanocrystalline Au sample, which was transformed to a 22.5° GB by GB dissociation induced twinning. **i** Summary of GB-associated deformation twinning in a wide range of FCC metallic films/foils, bulk samples and micro-crystals with different grain sizes[9,24,46–48,52–56]. Scale bars: **a–c** 2 nm; **d–f**, **g**, **h** 5 nm.

welding potential. In the current study, Au bicrystals containing [1$\bar{1}$0] tilt GBs (with GB misorientations $\theta = 23°$ and 87°) were successfully fabricated for tensile testing. During in situ experiments, the Au probe was moved backward slowly at a constant rate of 0.005 nm s$^{-1}$ to realize the tensile/shear loading, giving an estimated strain rate of 10$^{-3}$ s$^{-1}$. In all experiments, the TEM was operated at 300 kV with low current density to minimize the potential beam effect on deformation. The in situ deformation processes were recorded in real-time by a CCD camera at a rate of ~0.3 s per frame.

**MD simulations**. Atomistic simulations were performed to explore the microstructural origin of the twinning-assisted dynamic adjustment of GB mobility. The embedded atom method potential[50] used to compute the interatomic forces has been proven to be reliable in describing the fundamental properties of Au. The Au samples containing an inclined (Fig. 2)/flat (Fig. 3a, b) tilt GB was created by constructing two separate crystals with a designed crystallographic misorientation and joining them along the axial direction. To study the misorientation and inclination effects, a series of cylindrical bicrystal samples with GB misorientations ranging from 10° to 70°, and inclinations ranging from 0° to 90°, were created by rotating the upper grain G2 and the lower grain G1 (see the schematic diagrams in Fig. 4a, f). Each cylinder has a diameter of 16 nm and a height of 20 nm, containing a total of ~250,000 atoms. Three layers of atoms at the top and bottom boundaries of the cylinder were fixed as rigid slabs. The remaining dynamic atoms were allowed to adjust their positions in a Nose-Hoover thermostat at 300 K. Free boundary conditions were applied in all three directions of the cylindrical sample. The systems were relaxed for 20 ps to obtain equilibrated GB structures. The time step was chosen as 2 fs. A constant shear/tension velocity of $v = 1$ m s$^{-1}$ parallel/inclined to the boundary plane was applied on the rigid slab of the top grain. A velocity profile with a linear gradient from 0 to 1 m s$^{-1}$ was assigned to the dynamic atoms along the axial direction. Note that the shear and tension simulations at a strain rate of $5 \times 10^7$ s$^{-1}$ on the 23° GB (containing ~120,000 atoms) and with periodic boundary conditions to avoid the influence of free surfaces. In addition, the quasi-three-dimensional polycrystalline sample in Fig. 5g contained four hexagonal columnar grains with a grain size of 15 nm and a total of ~190,000 atoms. The grains are misoriented by the texture axis ⟨110⟩. Shear loading was applied with a constant strain rate of $5 \times 10^8$ s$^{-1}$. OVITO[51] was used to visualize the simulated samples, and the common neighbour analysis method was employed to determine the position and structural evolution of the GBs.

We determined the GB energy by calculating the average excess energies of the atoms at the GB compared with the energies possessed by normal FCC structured atoms, i.e.,

$$\gamma_{GB} = \frac{E_{total} - N \cdot E_{FCC}}{A} \quad (3)$$

where $E_{total}$ is the total potential energy of the simulated sample without free surface, $N$ is the total number of atoms used in the calculation, $A$ is the GB area, and $E_{FCC} = -3.924$ eV is the equilibrium energy of each FCC atom in a single crystalline Au lattice[50]. The atomic von Mises stress is defined as

$$\sigma_{von-Mises} = \sqrt{1/2[(\sigma_x - \sigma_y)^2 + (\sigma_x - \sigma_z)^2 + (\sigma_z - \sigma_y)^2 + 6(\tau_{xy}^2 + \tau_{yz}^2 + \tau_{xz}^2)]}$$

(4)

where $\sigma_x$, $\sigma_y$, $\sigma_z$, $\tau_{xy}$, $\tau_{yz}$, $\tau_{xz}$ are the six independent components of the per-atom stress tensor.

For NEB calculation of the GB migration barrier, we chose the initial state where the GB has been fully relaxed before each migration step and a final state where the GB has migrated for a distance of roughly one atomic layer (0.3 nm). The final state was minimized to set the system energy close to the corresponding initial states. And a series of replicas were created by linear interpolation to connect the two end-states. The activation energies were calculated by finding the minimum energy paths (MEP) and transition states of the migration process.

## Data availability
The data that support the findings of this study are presented in the paper and/or the Supplementary Materials.

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

## Acknowledgements

J.W. acknowledges the support of the National Natural Science Foundation of China (52071284 and 51771172) and the Innovation Fund of the Zhejiang Kechuang New Materials Research Institute (ZKN-18-Z02). H.Z. acknowledges financial support from the National Natural Science Foundation of China (11902289, 12172324) and computational support from the National Supercomputer Center in Beijing.

## Author contributions

J.W. proposed the idea. J.W. and H.Z. directed the project. Q.Z., Y.C. and J.W. conducted the experiments and analysed the data. Q.H. and H.Z. performed the simulations and analysed the data. Q.H., Q.Z., H.Z. and J.W. wrote the paper. M.G., J.L., W.Y., Z.Z. and J.W. contributed to the data analysis and the paper revision.

## Competing interests

The authors declare no competing interests.
