## [Peer Review File · Nature Communications]

REVIEWER COMMENTS

Reviewer #1 (Remarks to the Author):

It is a good paper devoted to the very important phenomenon, namely the influence of twinning on the deformation processes in fcc metals. It is because the nanocrystalline materials possess a large volume fraction of grain boundaries (GBs), which could substantially modify their physical, mechanical, and chemical properties in comparison with the coarse-grained polycrystalline counterparts. The paper could be published after major revisions. In my opinion, the authors underestimate the influence of GB faceting on the studied processes. It is good visible in Fig.1 that the twin GBs migrating under action of mechanical driving force are not continuously curved but contain various facets. This is a well-known fact (see for example Acta 56 (2008) 2728) and it is based on the strong dependence of GB energy on inclination of twin GBs. Moreover, the spectrum of crystallographically different facets (corresponding to the thermodynamic equilibrium) is different for different temperatures (see for example J. Mater. Sci. 51 (2016) 382). Close to the melting temperature, twin GB contain usually only two facets (symmetrical ones 111/111 and so-called 9R facets, Z.Metallk. 87 (1996) 911), but with decreasing temperature the number of crystallographically different facets in equilibrium increases. It is due to the crossing the faceting-roughening transition temperatures (Acta metall. 36(1988) 1573) for the facets with lower and lower specific density of coincidence sites. It can be supposed from the text that the in situ TEM experiments took place at room temperature (RT), which is very far away from melting temperature T_m of gold. At RT one can expect that the equilibrium spectrum of facets for twin GBs can contain at least six crystallographically different facets. The deformation-driven migration gives to these additional facets the chance to appear in the GB shape. This process is similar to the faceting of GBs slowly moving under the action of capillary driving force. I believe that the GB faceting-roughening transitions can definitely influence the phenomena observed and analyzed by the authors. Therefore, I would strongly propose to discuss these points in the paper.

Reviewer #2 (Remarks to the Author):

The main concern of this work is the dynamically self-adjust deformability of grain boundaries (GBs) under mechanical loading in metallic bicrystals using as integrated method of in situ high resolution transmission electron microscope (HRTEM) nanomechanical testing and atomistic simulations.

1) There is no physical reasoning behind decoupling shear and tension loading scenarios in Fig. 2. It seems to be that the authors decouple these loading scenarios for the sake of convenience, or were not able to have both loading components at the same time (even though there are elegant ways of doing combined loading). For the sake of discussion, it has been shown that having both tension and shear can result in the increase of generalized stacking fault energy of the material and impacting nanoscale plasticity phenomena in crystalline materials, such as dislocation dissociation, nucleation, and twinning: [R2] Andric, P., Yin, B. & Curtin, W. A. Stress-dependence of generalized

stacking fault energies. *Journal of the Mechanics and Physics of Solids* 122, 262–279 (2019). Therefore, decoupling shear and tension can result in activation of different deformation mechanisms.

2) Page 8, line 149: “noted that this twinning-assisted adjustment of GB deformability is a shear-driven process independent of free surfaces” The authors have provided no evidence to support this claim. This statement needs to be supported by additional MD simulations, with no decoupling of course, where GB is embedded in a bulk material without any free surfaces around. By the way, the dimensions are very small in this nanopillar. Therefore, the surface to volume ratio is notably large, and the free surfaces can have notable impacts on all types of deformation mechanisms here.

3) It is not clear how the authors calculated the elastic energies for the nanopillars. Especially, using rigid walls at the top and bottom of a pillar can make the whole pillar notably stiffer than it should be and result in artifacts in elastic constant calculations. Also, the authors totally ignored thermally activated mechanisms in the definition of the driving force.

4) How did the authors calculate per atom stress in the supplementary figure 4? Where are the details if someone wants to reproduce those results?

5) As it was mentioned before, freezing atoms in the three layers at the top and bottom can severely constrain the system. Most importantly, the authors used 300 K relaxation in their MD for 20 ps which does not do anything in finding configurations with minimum energy due to very limited timescale that can be accessed by MD. This situation becomes worse due to constraining the system by the rigid walls, and the grains do not have any chance to adjust themselves at the grain boundary to relax the stresses at the boundary. Therefore, what the authors loaded is not properly relaxed, and those initial configurations are nothing but simply equilibrated configurations at 300 K. By the way, thermostating the whole system and manipulating the atoms to induce linear shear strain through the height of the pillars can notably interfere with the atomic trajectories and generate subsequent spurious effects. By the way, it does not seem to be physically meaningful to thermostat the whole pillars where there are free surfaces everywhere. The system should be able to maintain the 300 K temperature if properly relaxed and loaded.

This work is recommended for publication with major revisions.

Point-by-point responses to review comments

Manuscript Ref: NCOMMS-21-20198

Title: Twinning-assisted dynamic adjustment of grain boundary mobility

Authors: Qishan Huang, Qi Zhu, Yingbin Chen, Mingyu Gong, Jixue Li, Ze Zhang, Wei Yang, Jian Wang, Haofei Zhou, Jiangwei Wang

We sincerely thank the reviewers for their careful review of our manuscript and valuable comments/suggestions on our work. In the following, the review comments are listed in *italic font* and our response to each comment is given in blue text. The manuscript has been carefully revised accordingly.

Reviewer #1 (Remarks to the Author):

It is a good paper devoted to the very important phenomenon, namely the influence of twinning on the deformation processes in fcc metals. It is because the nanocrystalline materials possess a large volume fraction of grain boundaries (GBs), which could substantially modify their physical, mechanical, and chemical properties in comparison with the coarse-grained polycrystalline counterparts. The paper could be published after major revisions.

Response: We thank the reviewer for the positive comments on the scientific significance of our work.

In my opinion, the authors underestimate the influence of GB faceting on the studied processes. It is good visible in Fig.1 that the twin GBs migrating under action of mechanical driving force are not continuously curved but contain various facets. This is a well-known fact (see for example Acta 56 (2008) 2728) and it is based on the strong dependence of GB energy on inclination of twin GBs. Moreover, the spectrum of crystallographically different facets (corresponding to the thermodynamic equilibrium) is different for different temperatures (see for example J. Mater. Sci. 51 (2016) 382). Close to the melting temperature, twin GB contain usually only two facets (symmetri ones 111/111 and so-called 9R facetes, Z. Metallk. 87 (1996) 911), but with dereasing temperature the number of crystallographically different facets in equilibrium increases. It is due to the crossing the faceting-roughening transition temperatures (Acta metall.36(1988) 1573) for the facets with lower and lower specific density of coincidence sites. It can be supposed from the text that the in situ TEM experiments took place at room temperature (RT), which is very far away from melting temperature T_m of gold. At RT one can expect that the equilibrium spectrum of facets for twin GBs can contain at least six crystallographically different facets. The deformation-driven migration gives to these addotional facets the chance to appear in the GB shape. This process is similar to the faceting of GBs slowly moving under the action of capillary driving force. I believe that the GB faceting-roughening

transitions can definitely influence the phenomena observed and analyzed by the authors. Therefore, I would strongly propose to discuss these points in the paper.

Response: We thank the reviewer for the insightful comments and helpful references/suggestions on GB faceting. We agree that in our experiment, the twinned GBs migrating under the action of mechanical driving force are not continuously curved but contain various facets. In a crystallographic sense, the influence of GB facets can be treated as the impact of GB inclination φ (*Acta Mater.* **56**, 2728–2734 (2008); *Phys. Rev. B* **4**, 113402 (2020)) according to the macroscopic geometric degree of freedoms. The GB inclination φ affects both the twinning behaviors at the GB and the GB migration afterwards, which will be respectively discussed as follows:

(i) Faceting effects on the twinning at GB. We would like to stress that GB inclination is indeed of general significance to twinning at the GB, which has been shown by the twinning tendency map in Fig. 4f of the revised manuscript. For the specific case in Fig. 1 and Fig. 2d, the main GB facet (A) with $\theta=23^\circ$ and $\varphi=66^\circ$ on the pristine GB possesses an intrinsically high twinning tendency (TT=1.28), compared with other minor facets B and C with $\varphi=36^\circ$ (TT=0.53) and 16° (TT=0.38), respectively. Therefore, twinning is predicted to occur preferentially at the GB facet A, which is consistent with our *in situ* TEM observation of twin nucleation and continuous growth along the main GB facet A (as shown below in Fig. R1). The minor GB facets B and C, however, were removed during GB migration via the sequential lateral motion of these GB ledges. To better support the inclination effects on twinning, we have supplemented additional MD simulations regarding the deformation of $\theta=47^\circ$ GB with varying inclinations in Fig. R2 and $\theta=80^\circ$ GB with varying inclinations in Supplementary Fig. 6 of the revised Supplementary Information.

Fig. R1 Preferential twin nucleation and growth along the GB facet A with a twinning tendency of TT=1.28, considering the collective influences of its misorientation $\theta=23^\circ$ and inclination $\varphi=66^\circ$.

Fig. R2 The inclination effects on twinning of $\theta=47^\circ$ GB. (a-b) Twinning was activated with high twinning tendency ($TT>0.93$). (c-d) Twinning was absent for a GB with insufficient twinning tendency ($TT = 0.82$).

(ii) Faceting effects on GB migration. Due to the heterogeneous nucleation and twin growth at the GB, the GB_{T-2} (after full twinning) exhibits a binary configuration, which includes a curved segment with various facets (near the bottom surface) and a flat segment with relatively lower boundary energy (near the upper surface). For the curved segment, we agree with the reviewer that various facets are contained, which either shrink or expand during the GB migration, depending on the respective energy and mobility of each facet. Therefore, the facet-roughening transition processes (pointed out by the reviewer) should play a critical role during the GB migration, which is confirmed by the topological transition from a faceted GB to a comparatively flat GB with significantly reduced faceting (Fig. 1e-f). We have cited the related references mentioned by the reviewer about the faceting-roughening transition in the revised manuscript to enrich our discussion on the GB migration behaviors after twinning.

In addition, we notice that this energy-driven facet-roughening transition may impose certain (positive) dragging force on the flat segment, which has been confirmed by our experimental observations of a much larger migration distance of the flat GB segment compared with that of the faceted segment (Fig. R3). Yet, we need to point out that although the two GB segments may differ in their dynamic behaviors under mechanical loading, the average migration rate of the whole GB rises significantly compared with that of the original GB before twinning, which confirms the dominant role of GB twinning in adjusting the GB migration rate.

Fig. R3 Superimposed GB profiles showing the migration distances of different GB segments after twinning.

Therefore, we have re-emphasized the physical background of the (misorientation and inclination-governed) twinning tendency map (Fig. 4f), added the above discussion about the influences of GB faceting (Fig. 2d-f) in the revised manuscript, following the reviewer's valuable suggestions.

Reviewer #2 (Remarks to the Author):

The main concern of this work is the dynamically self-adjust deformability of grain boundaries (GBs) under mechanical loading in metallic bicrystals using as integrated method of in situ high resolution transmission electron microscope (HRTEM) nanomechanical testing and atomistic simulations.

Response: Thanks for reviewer's valuable and helpful comments on our manuscript.

1) There is no physical reasoning behind decoupling shear and tension loading scenarios in Fig. 2. It seems to be that the authors decouple these loading scenarios for the sake of convenience, or were not be able to have both loading components at the same time (even though there are elegant ways of doing combined loading). For the sake of discussion, it has been shown that having both tension and shear can result in the increase of generalized stacking fault energy of the material and impacting nanoscale plasticity phenomena in crystalline materials, such as dislocation dissociation, nucleation, and twinning: [R2] Andric, P., Yin, B. & Curtin, W. A. Stress-dependence of generalized stacking fault energies. *Journal of the Mechanics and Physics of Solids* 122, 262–279 (2019). Therefore, decoupling shear and tension can result in activation of different deformation mechanisms.

Response: Thanks for reviewer's comment on the influence of combined tensile and shear stresses on deformation mechanism. We fully agree with the reviewer that both tension and shear can result in the increase of generalized stacking fault energy of the material and impacting nanoscale plasticity phenomena in crystalline materials, which should be careful when decoupling the shear and tension loading. To address this comment, we have now performed an additional simulation on a sample with an inclined 23° GB identical to that of our experiment (Fig. 1a) to investigate the dynamic response of the GB under uniaxial tension (Fig. R4a). This inclined GB is apparently subjected to a combined tensile and shear stress state. During deformation, the GB was decomposed into embryonic deformation twins (Fig. R4b), followed by subsequent twin formation and connection (Fig. R4c), leading to the twinning-assisted GB structure adjustment (Fig. R4d). These observations are consistent with those observed in our TEM experiments (Fig. 1a-f) and previous MD simulations (Fig. 3a).

Fig. R4 Twinning assisted GB adjustment of the inclined GB_{1-2} under tension. Scale bar: 5 nm.

It is noted that dislocation nucleation from the free surface is inevitable with the straining of above simulation, which can influence the further migration of GB_{T-2} (Fig. R4c-d). Given that twinning is a process dominated by the shear stress, we therefore

decouple the shear and tension loading by setting up a simulation model with a horizontal 23° GB in Fig. 3, to avoid the influence of surface nucleation on GB development. The model has a periodic boundary condition along the GB plane to get rid of the free surfaces (Fig. 3). To further validate this model, an additional simulation with combined tension and shear was performed on this model, following the reviewer’s suggestion (Fig. R5). Specifically, the shear loading was applied parallel to the GB with a strain rate of $5 \times 10^7 \text{ s}^{-1}$, while the tension loading was applied perpendicular to the GB with a strain rate of $2.5 \times 10^6 \text{ s}^{-1}$ (Fig. R5b). The ratio of the shear and tension strain rates was chosen to be 20, based on the inclination angle between the inclined GB and the tensile loading direction in Fig. 1a. Similar twinning behavior was also observed under this combined shear and tension simulation of horizontal GB, consistent with the results from decoupled shear simulation (Fig. 3a) and the uniaxial tension simulation on inclined GB (Fig. R4). We have also investigated the effect of tension loading on the deformation of the horizontal GB (Fig. 3b), which merely show the occasional dislocation nucleation from the GB, without any noticeable GB migration or twinning. These simulation results indicate that the twinning-assisted self-adjustment of the 23° GB is a shear-dominated process and the tensile component may not be a governing factor.

Fig. R5 Coupled tension and shear loading induced twinning and GB dynamic adjustment. (a) Initial configuration of a flat 23° GB with periodic boundary conditions along GB plane. (b-c) TB segments nucleation and connection under coupled shear and tension. (d) Migration of the newly formed GB_{T-2} migrated under combined loading. Scale bar: 2 nm.

To further clarify this critical question, we have added above discussions and the corresponding simulation results in the revised manuscript, following the reviewer’s suggestion.

2) Page 8, line 149: “noted that this twinning-assisted adjustment of GB deformability is a shear-driven process independent of free surfaces” The authors have provided no evidence to support this claim. This statement needs to be supported by additional MD simulations, with no decoupling of course, where GB is embedded in a bulk material without any free surfaces around. By the way, the dimensions are

very small in this nanopillar. Therefore, the surface to volume ratio is notably large, and the free surfaces can have notable impacts on all types of deformation mechanisms here.

Response: We thank the reviewer for this comment. First, two experimental examples with GBs embedded in nanocrystalline samples (Fig. 5a-f) showed the twinning-assisted GB adjustment process, indicating that it can be activated without the presence of free surfaces. Second, we have performed an additional MD simulation on a polycrystalline sample with the GB embedded in a bulk and periodic boundary conditions are imposed to all directions to avoid the influence from free surfaces, as recommended by the reviewer. Besides, a combined shear and tension loading was applied on this sample. The simulation results shown in Fig. R6 rule out the dependence of the twinning-mediated GB deformation on free surface. Third, we have also performed a simulation on a horizontal GB with periodic boundary conditions applied to all three directions to mimic the bulk environment (Fig. R7). The simulation results further verify our claim. As a result, our experimental evidences along with additional MD simulations have demonstrated that the dynamic GB adjustment behavior is a process independent of free surfaces.

We have added these results and discussions in the revised Supplementary Information.

Fig. R6 Twinning-assisted Deformation dynamics of an 87° GB in a polycrystalline Au sample with coupled tension and shear loading. Scale bar: 5 nm.

Fig. R7 Periodic boundary conditions applied in all three directions of the sample, illustrating the twinning-assisted deformation dynamics of the 23° GB in Au with coupled tension and shear loading. Scale bar: 2 nm.

3) It is not clear how the authors calculated the elastic energies for the nanopillars. Especially, using rigid walls at the top and bottom of a pillar can make the whole pillar notably stiffer than it should be and result in artifacts in elastic constant calculations. Also, the authors totally ignored thermally activated mechanisms in the definition of the driving force.

Response: We thank the reviewer for these constructive comments. First, we would like to point out that the elastic energies of G1, G2 and T were not obtained from MD simulations. They were obtained by theoretical calculations using the Hooke's law (Supplementary Discussion 2), given that a uniaxial tension loading was applied to the bicrystal with an inclined GB (Fig. R8). The energetic driving force P for GB migration can be estimated as the energy gradient across the GB, which is commonly known as the Eshelby force in solid mechanics (*J. Elast.* **5**, 321-335 (1975), *J. Appl. Mech.-Trans. ASME*, **67**, 829–830 (2000)). In principle, Eshelby force arises from the release of stored elastic strain energy of a system associated with the movement of lattice defects (including GB) in the system. We have therefore adopted it as the energetic driving force to understand the variation in GB migration rate before and after twinning.

Fig. R8 Schematic diagram shows theoretical calculation of the driving force of GB migration. F_{\pm} represents the elastic energy densities of the top and bottom grains subjected to an axial loading, and $[C]$ represents the elastic constant matrix of the grains.

We fully agree with the reviewer that thermal activation is important to understand the driving force of GB migration. At finite temperatures, thermal activation permits the migration mechanisms of GB_{1-2} and GB_{T-2} , which is assisted by the distortion and transformation of GB structural units. Quantitatively, we have performed additional simulations to calculate the energy barrier for thermal activation of GB migration. Specifically, we employed the nudged elastic band (NEB) method (*Proc. Natl Acad. Sci. USA* **104**, 3031–3036 (2007); *J. Chem. Phys.* **113**, 9901–9904 (2000)) to determine the energy barrier of GB migration by thermal activation. The energy barrier (activation energy) is defined as the energy difference between the

saddle-point and initial state on the minimum energy path (MEP) of GB migration. In the NEB calculation, two end-states (initial and final states) were first determined, then a discrete elastic band consisting of a finite number of replicas (images) of the system was constructed by linear interpolation to connect the two end-states. With appropriate relaxation, the band converges to the MEP. Starting from the initial GB structure, we have performed 4 consecutive migration steps along the GB normal direction for both GB_{T-2} and GB₁₋₂ (see the insets of Fig. R9). In each unit step, the GB migrated upwards with a distance of 0.3 nm (roughly equals to one-atomic-layer spacing). The migration energy barrier of GB_{T-2} averaged from these consecutive unit migration steps (cyan dotted line in Fig. R9) is much lower than that of GB₁₋₂ (red dotted line in Fig. R9), further supporting the twinning-enhanced GB migration rate in the present study.

We have added these additional results and discussions on thermal activation of GB migration in the revised manuscript.

Fig. R9 Comparison between the thermal migration barriers of GB₁₋₂ and GB_{T-2} measured from 4 consecutive migration steps along the GB normal direction, with a migration distance of one atomic layer (0.3 nm) in each unit step. The averaged values of migration barriers of GB₁₋₂ and GB_{T-2} from these migration steps are highlighted by the red and cyan dotted lines, respectively.

4) How did the authors calculate per atom stress in the supplementary figure 4? Where are the details if someone wants to reproduce those results?

Response: The atomic von Mises stress in previous supplementary figure 4 (current supplementary figure 5) is defined as:

$$\sigma_{von-Mises} = \sqrt{1/2[(\sigma_x - \sigma_y)^2 + (\sigma_x - \sigma_z)^2 + (\sigma_z - \sigma_y)^2 + 6(\tau_{xy}^2 + \tau_{yz}^2 + \tau_{xz}^2)]}$$

where σ_x , σ_y , σ_z , τ_{xy} , τ_{yz} , τ_{xz} are the six independent components of the per-atom stress tensor. The atomic von Mises stress contour was used to visualize the stress concentration at the GB and evaluate the plastic deformation in MD simulations. The

relevant simulation details have now been provided in the Methods in the revised manuscript.

5) As it was mentioned before, freezing atoms in the three layers at the top and bottom can severely constrain the system.

Response: Freezing atoms in the three layers at the top and bottom is to load the system. Practically, to load the sample in MD simulations, force and velocity of the boundary atoms should set to be zero. This is a standard boundary condition typically used in stress-driven simulations, while free boundary conditions were always used in random walk simulations. How strong the constrain effect of boundary condition would depend on the height of the model and loading condition. Corresponding to the GB-mediated deformation in our work, all deformation events were activated along the interface, while the boundary hardly affect the simulation result. For the tension simulation, the total strain was less than 0.35, which did not generate a significant shrinkage of the model in the horizontal direction, so no plastic events were activated at the interface between the fixed and unfixed atoms.

Most importantly, the authors used 300 K relaxation in their MD for 20 ps which does not do anything in finding configurations with minimum energy due to very limited timescale that can be accessed by MD. This situation becomes worse due to constraining the system by the rigid walls, and the grains do not have any chance to adjust themselves at the grain boundary to relax the stresses at the boundary. Therefore, what the authors loaded is not properly relaxed, and those initial configurations are nothing but simply equilibrated configurations at 300 K.

Response: We agree that GB relaxation is indeed an important question. We have performed a series of simulations at two different temperatures (300 K and 600 K) for various relaxation times ranging from 10 ps-80 ps (Fig. R10). It shows that GB relaxation at 300 K for 20 ps seems to be sufficient to minimize the GB energy. Increasing the relaxation time and temperature led to negligible energy fluctuation (Fig. R10a) and structure difference of the GBs (Fig. R10b).

Fig. R10 Relaxation time effect on GB energy and GB structure. (a) Relaxation time

related GB energy fluctuation. GB energy was calculated after different relaxation times and energy minimizations. (b) GB structures corresponding to different relaxation times.

Furthermore, we have also created a series of initial GB structures by shifting the top and bottom grains relatively for 0-2 Å along x direction and 0-2 Å along y direction within the GB plane. The GB energy obtained after different relaxation times at 300 K are presented in Fig. R11a-b. We found that the initial GB structure has imperceptible influence on the final GB energy, as long as sufficient relaxation time was applied.

Finally, we examined the deformation behaviors of these GBs. Fig. R11c-e shows the deformation patterns of GBs with either different relaxation times (0-50 ps) or different initial structures (shifted along x and y directions within the GB plane). We found that the mechanism of twinning-mediated GB deformation was activated in all cases. These results clearly indicate that the relaxation and preparation of GBs in our MD simulations can capture the experimentally observed GB deformation mechanism.

Fig. R11 Initial GB structure impact on GB energy and deformation. Imperceptible difference in GB energy after slightly shifting the initial GB structure in (a) x and (b) y directions within the GB plane. (c) Twinning-mediated GB deformation under shear loading at 300 K in samples relaxed for different time periods of $t=0$ ps, 10 ps, 20 ps, 30 ps, 40 ps, 50 ps. (d-e) The same mechanism was also observed in the shifted GBs under shear loading, which were shifted for a distance of 0.5 \AA along x direction ($dx=0.5 \text{ \AA}$) and a distance of 0.5 \AA along y direction ($dy=0.5 \text{ \AA}$), respectively. All of the shifted GBs were relaxed for 20 ps before shear loading. Scale bars: 2 nm.

By the way, thermostating the whole system and manipulating the atoms to induce linear shear strain through the height of the pillars can notably interfere with the atomic trajectories and generate subsequent spurious effects. By the way, it does not seem to be physically meaningful to thermostat the whole pillars where there are free surfaces everywhere. The system should be able to maintain the 300 K temperature if properly relaxed and loaded.

Response: In MD simulations, there are two typical ways to achieve a certain strain. One way is to apply deformation gradients increment to the whole system, the other way is to continuously apply displacements in a certain region. Both approaches have advantages and shortcomings. In our simulations, we adopted a constant displacement increment in certain region, which generates constant straining to the simulation system. The fixed atoms do not participate in MD simulations and only serve to impose interatomic forces on neighboring dynamic atoms. And introducing initial linear shear velocity profile was to reduce equilibrium time and velocity fluctuations of atoms, which effectively prevents stress wave induced by instantaneous velocity gradient.

Regarding temperature control, we agree that thermostating will affect the motion of atoms especially for that located in the defected regions. More strictly

speaking, thermostating often reduces the possibility of unlikely activated events (such as atoms diffusion) because atoms with extra high kinetic energy will be quenched to reduce their thermal energy. Corresponding to our work, such a thermostating does not affect the simulation results, because atoms diffusion is not the key process.

REVIEWERS' COMMENTS

Reviewer #1 (Remarks to the Author):

After revision the paper is acceptable for publication as it is

Reviewer #2 (Remarks to the Author):

The authors have adequately addressed the reviewer's concerns. The paper is accepted for publication in its current form.